# Differential Diagnosis between Child Abuse and Infantile Cortical Hyperostosis: A Case Report and Literature Review

**DOI:** 10.3390/ijerph182212269

**Published:** 2021-11-22

**Authors:** Du-Yeon Lee, Woo-Jong Kim, Byungsung Kim, Jae-Hwi Nho, Chang-Hwa Hong, Sang-Mi Lee, Ik-Dong Yoo, Changeui Lee, Ki-Jin Jung

**Affiliations:** 1Department of Orthopaedic Surgery, Soonchunhyang University Hospital Cheonan, 31 Suncheonhyang 6-gil, Dongnam-gu, Cheonan-si 31151, Korea; keyong2271@naver.com (D.-Y.L.); kwj9383@hanmail.net (W.-J.K.); chhong@schmc.ac.kr (C.-H.H.); 129840@schmc.ac.kr (C.L.); 2Department of Orthopaedic Surgery, Soonchunhyang University Hospital Bucheon, 170, Jomaru-ro, Bucheon-si 14584, Korea; kbsos@schmc.ac.kr; 3Department of Orthopaedic Surgery, Soonchunhyang University Hospital Seoul, 59, Daesagwan-ro, Yongsan-gu, Seoul 04401, Korea; huuytime@gmail.com; 4Department of Nuclear Medicine, Soonchunhyang University Hospital Cheonan, 31 Suncheonhyang 6-gil, Dongnam-gu, Cheonan-si 31151, Korea; c91300@schmc.ac.kr (S.-M.L.); 92132@schmc.ac.kr (I.-D.Y.)

**Keywords:** child abuse, hyperostosis, cortical, congenital

## Abstract

Child abuse is a major public health problem that can lead to critical consequences for the child and family. However, early identification of abuse may be difficult. An 8-month-old boy presented with extensive periosteal reaction in both upper and lower long bones. There was no specific history of injury. Caffey disease was initially considered as the diagnosis because the patient displayed fever and hyperostosis of multiple bones with elevated erythrocyte sedimentation rates and C-reactive protein and alkaline phosphatase levels. However, we suspected child abuse based on the clinical and radiological features. We eventually found out that the child had been injured through child abuse and were able to treat him. We report this case because child abuse cases may be confused with Caffey disease. This case report can, therefore, help distinguish between Caffey disease and child abuse.

## 1. Background

Among children younger than 15 years, the World Health Organization estimates that child abuse accounts for 13% of the 1.2 million deaths due to injury worldwide. In the United States, up to 2500 children die of inflicted injuries annually, with children under one year of age being affected disproportionately [1]. Child abuse is a major public health problem to the extent that even this incidence is thought to be underreported. Although attention toward child abuse and the severity of action against it are increasing, it remains challenging to diagnose. Case histories can be provided incorrectly or not be provided intentionally by caretakers, which may mislead the clinician to miss the diagnosis of child maltreatment. Cutaneous injury, known as the most common injury of abuse, can sometimes fade, despite the presence of significant skeletal injury [2]. Although early identification of abuse is difficult, it is important to detect cases of minor injury to protect the child and family from future harm. The important role of the orthopedic specialist is also emphasized in a mini review by Pavone et al. Therefore, health care providers should be familiar with signs of child maltreatment and the medical conditions that may mimic child abuse. This study was published to highlight the importance of the differential diagnosis between Caffey disease and child abuse.

Prior to the publication of Dr. John Caffey’s 1946 article on the association between chronic subdural hematoma and long bone fractures in infants, child abuse was essentially unrecognized by the medical community [3]. In addition to establishing the pediatric skeletal anatomy in child abuse and its normal variance, Caffey identified many skeletal abnormalities, including infantile cortical hyperostosis, which is now known as Caffey disease [4].

Caffey disease is a rare condition characterized by massive hyperostosis with fever, soft tissue swelling, and pain. Caffey disease may resemble a fracture with cortical hyperostosis and should be ruled out along with a few other metabolic diseases, prior to determining a diagnosis of child abuse. However, Caffey disease or metabolic diseases are very uncommon diagnoses among children evaluated for abuse. For example, in a prospective observational study of 2890 children undergoing evaluation for physical abuse, only 19 children had metabolic bone disease including vitamin D deficiency, osteoporosis, hyperparathyroidism, and Menkes syndrome [5]. Children with metabolic bone disease may have pathological fractures induced by minor trauma, which may seem trivial compared to the skeletal injuries of abused children. Distinguishing metabolic bone disease from child abuse includes appropriate laboratory and radiological examination.

However, we report a very ambiguous case that may have been misdiagnosed as a systemic disease rather than a fracture due to a symmetric periosteal reaction of the whole body. Herein, we report a child abuse case with multiple fractures serious enough to suspect a systemic condition such as Caffey disease.

## 2. Case Description

This case report was approved by the International Review Board of Soonchunhyang University Hospital. The patient’s current guardian gave written informed consent for publication of this case report and accompanying images. An 8-month-old boy presented with bilateral severe chemosis, corneal opacity, and corneal ulcer. He was diagnosed with seborrheic dermatitis one month previously and a moisturizer had been used to treat it. According to the mother’s report, he had pruritus and hyperemia three days before the hospital visit, had been rubbing his eyes, and had not tried to properly open his eyes for two days prior to the visit.

An examination revealed bilateral purulent eye discharge and severe corneal erosion with opacity, which made it impossible to identify the anterior chamber (Figure 1). It is clinically rare for such a condition to have progressed to this extent in only three days. There was eschar on both canthal areas, which, according to his mother, was produced by rubbing but appeared to be more likely due to severe trauma.

He was born at full term by a cesarean section without any abnormalities on the newborn screening test, and his birth weight was 3.3 kg. He was an only child, looked well-nourished, and his immunization status was up to date.

At the time of the visit, he had a fever of 38.1 °C. In an initial blood examination, his platelet count was 521 × $10^{9}$/L, white blood cell count was 22.45 × $10^{9}$/L, Neutrophil count was 16.42 × $10^{9}$/L, erythrocyte sedimentation rate (ESR) was 61 mm/h, alkaline phosphatase (ALP) was 493 IU/L (reference range, 39–117 IU/L), and C-reactive protein (CRP) was 55.23 mg/L (reference range, 0–5.0 mg/L).

Considering the discrepancy between the history provided and the clinical findings, we obtained a skeletal survey to assess possible child maltreatment.

Skin lesions such as bruises or abrasions were not observed in the extremities, but an extensive periosteal reaction with transverse fractures was observed in both upper and lower long bones (Figure 2). There was no specific history of injury. Radiographs showed multiple fractures of both femurs, right tibia, and both ulna in various stages of healing. Spiral fractures were observed in the right tibia and right ulna (Figure 3).

Caffey disease was initially considered as the diagnosis because the patient displayed fever and hyperostosis of multiple bones with elevated ESR, CRP, and ALP levels. With further evaluation, however, Caffey disease was excluded, considering several findings; the infant’s flat bones, such as the mandible, skull, and ribs, which are commonly affected by Caffey disease, were intact. Additionally, in this case, soft tissue swelling and inflammation around the affected bones were not observed. Furthermore, the hyperostosis observed was not limited to the diaphysis and included the metaphysis and epiphysis, accompanying cortical disruption of the actual fracture, which is unlikely in Caffey disease.

Therefore, we suspected child abuse based on the following clinical and radiological features: multiple fractures present in an infant under 18 months of age without any history of major trauma, femoral fracture in a nonambulatory infant, and hyperostosis involving epiphysis rather than diaphysis. The patient was found to be a victim of child abuse through a police investigation, and three months have passed since then, and the fracture has improved (Figure 4). Because it was concluded that it was parental child abuse, we learned that child abuse should not be excluded from the visible parental actions of caring for a child.

## 3. Discussion

The Child Abuse Prevention and Treatment Act defines the maltreatment of a child as “any recent act or failure to act on the part of a parent or caretaker, which results in death, serious physical or emotional harm, sexual abuse or exploitation, which presents an imminent risk of serious harm”. Lately, public recognition of child abuse and neglect has improved considerably, but unfortunately, the issue remains common and serious. Investigators reveal that child abuse and neglect is a major public health problem resulting in critical health sequelae for the affected child.

We report the case of an 8-month-old, male child with periosteal reaction of bone in the whole body; therefore, Caffey disease was initially suspected. However, through precise analysis of the radiology, we considered that it was close to a fracture. Radiologically, periosteal thickening was seen over multiple bones, but the patterns differed from those of Caffey disease. First, unlike the bone lesions of Caffey disease that characteristically involve the mandible and skull, no flat bone involvement was evident in our case. In Caffey disease, the mandible is most frequently involved (in 70–90% of the cases) and is often used as a defining diagnostic criterion [6,7]. Second, in Caffey disease, a diaphyseal involvement with sparing of the epiphysis is common; however, the periosteal thickening observed in this case was at the epiphysis and metaphysis. Third and most importantly, there were cortical disruptions of bone indicating a fracture. The emphasis of this report is that Caffey disease is an important differential diagnosis for child abuse and clinicians must accurately identify its characteristics. Laboratory findings in affected patients may include elevated ESR, CRP and ALP levels; however, since this patient had concurrent eye inflammation, elevated ESR and CRP levels did not act as a differentiating factor.

Clinical features that should be considered for abuse include fractures with no history of major trauma, fractures in nonambulatory infants, multiple fractures, inconsistent history conflicting with the child’s development, changing history when asked repeatedly, or unexpected delay in seeking medical care. Abusive skeletal injuries are more common in infants and young children than in older children due to their small differential size relative to the perpetrator. In addition, since infants have relatively less mobility, accidental fractures are uncommon among them. Worlock and colleagues concluded that 80% of the fractures occurring in children aged 18 months or younger are abusive fractures, whereas only 2% are accidental [8]. These fractures may have no external sign of trauma; therefore, a skeletal survey must be performed for a child suspected of receiving abuse. A skeletal survey is a series of approximately 20 radiographs including those of the skull, long bones, hands, feet, thorax, pelvis, and spine. The guidelines of the American Academy of Pediatrics recommend that a skeletal survey should be carried out in all children less than two years old who are suspected of being abused. Skeletal surveys may also be important for older children with disabilities who are unable to disclose abuse and have a high incidence of maltreatment [9].

Dating fractures is important for assessing abusive skeletal injuries. Discrepancies between recorded history and the date of fracture suggested by imaging may support the diagnosis of abuse, as does the finding of multiple fractures in different stages of healing [10]. While dating a fracture, metaphysis and diaphysis may need to be assessed differently because their healing processes differ. Other types of imaging, for example, bone scans, can be performed to detect subtle fractures of rib, spine, and scapula.

There are several medical conditions that may resemble the findings of abusive skeletal injury and may need to be ruled out before the diagnosis of child abuse. Differential diagnosis may need to be performed based on laboratory or other medical evidence for forensic reasons, even if these conditions can be clinically excluded.

Osteogenesis imperfecta (OI), one of the major differential diagnoses of child abuse, is a genetic disorder of collagen formation that results in brittle bones. OI patients easily break bones, even with minimal trauma. Because OI is described as causing vulnerability to bruises, subdural hemorrhage, and retinal hemorrhage, it is the prototype for diseases that may be confused with child abuse [11,12], but is much less common than child abuse. Diagnosis of OI is usually performed based on the family history, physical examination, and radiological findings. Indeed, OI is extremely rare (<3 cases per 1,000,000 population) in children suspected of being abused [13]. Other medical conditions to be considered include osteopenia of prematurity, rickets, scurvy, secondary hyperparathyroidism, Menkes kinky hair syndrome, as well as some skeletal dysplasias, malignancies, neuromuscular disorders or other diseases that result in osteopenia because of limited mobility [14].

In this case, we needed to differentiate the conditions from Caffey disease. Caffey disease, also known as infantile cortical hyperostosis, is an inflammatory disease characterized by fever (sometimes as high as 40 °C [104 °F]), irritability, subperiosteal bone hyperplasia, soft tissue swelling, and pain adjacent to the involved bones. The bone changes typically begin between birth and five months of age and resolve by two years. Radiological findings of subperiosteal cortical hyperostosis typically involve the mandible, long bones, clavicles, scapulae, and ribs; however, any bone may be involved. The pattern of long bone invasion includes the diaphysis, with sparing of the epiphysis. Mandibular involvement is seen in 70–90% of the cases and can be useful to differentiate Caffey disease from child abuse.

However, in the present case, neither the mandible nor the skull were invaded, there was a tendency for epiphyseal rather than diaphyseal invasion of long bones, and extensive cortical disruption; therefore, we diagnosed the findings as child abuse-related fractures. Although rib fractures and abdominal injuries were also suspected after expert inspections, no abnormalities were found. There were, however, multiple fractures, including clavicle fractures, that were identified, but all were possible to manage with conservative treatment. Conservative treatment using a simple splint was thus performed, and complete union and remodeling findings were observed after 3 months. The patient is now isolated from his parents and is being treated in a shelter. The three-month follow-up was the last at our hospital. Our case study, therefore, emphasizes that clinicians must obtain a detailed history and perform a thorough examination. Before diagnosing a child of maltreatment, medical conditions that resemble child abuse should be fully evaluated. However, the presence of metabolic bone disease may not exclude the possibility of concurrent child abuse. The clinician’s role is not only in prevention and early detection but also in continuing to care for and manage the child and the family.

## 4. Conclusions

We report this case because child abuse cases may be confused with Caffey disease. This case report can, therefore, help distinguish between Caffey disease and child abuse.

## Figures and Tables

**Figure 1 ijerph-18-12269-f001:**
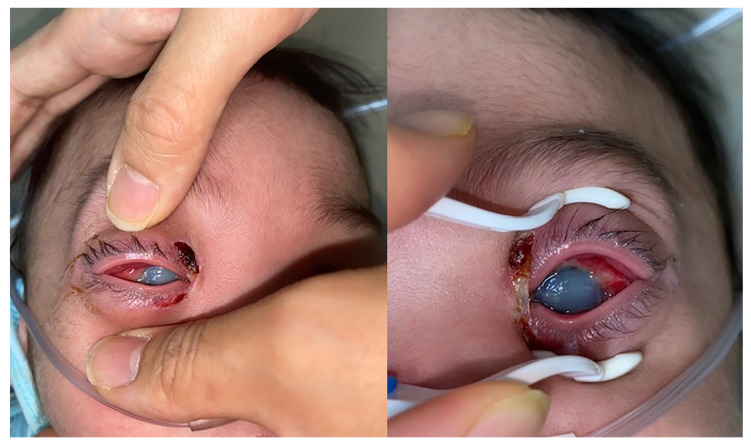
Anterior photography showing chemosis, corneal opacity, and corneal ulcer.

**Figure 2 ijerph-18-12269-f002:**
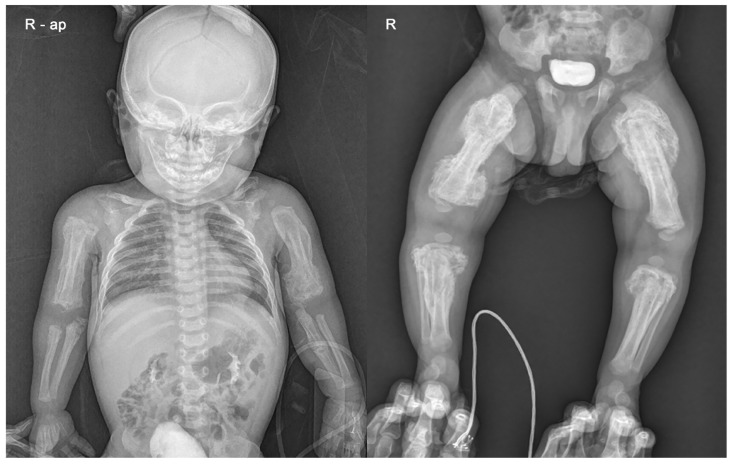
Infantogram and radiograph showing extensive periosteal reaction with multiple fractures in both upper and lower long bones.

**Figure 3 ijerph-18-12269-f003:**
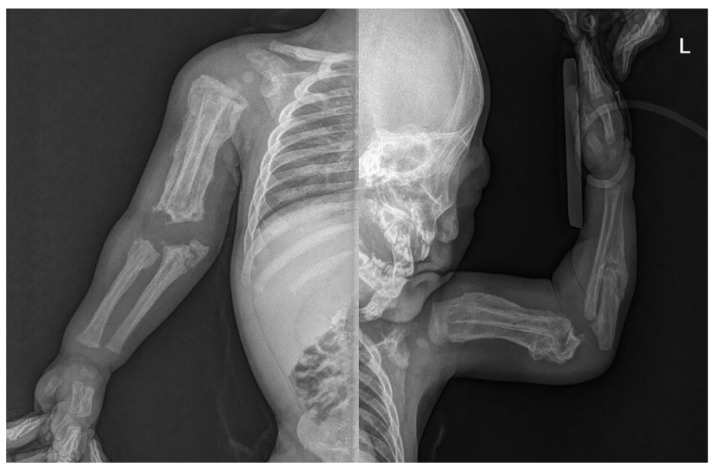
Radiograph of upper limb showing hyperostosis with fractures of both ulna.

**Figure 4 ijerph-18-12269-f004:**
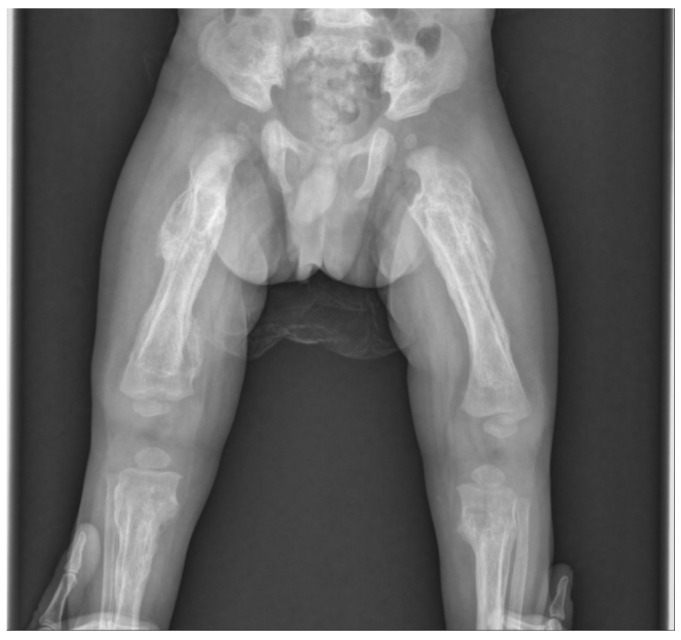
After 3 months, radiographical findings showed that the fracture was healed and remodeled.

## Data Availability

Data sharing is not applicable to this article. No new data were created or analyzed in this study.

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
