# Peer review of "Differential Diagnosis between Child Abuse and Infantile Cortical Hyperostosis: A Case Report and Literature Review"

_ijerph, 2021, doi:10.3390/ijerph182212269_

Round 1

Reviewer 1 Report

Thank you for reviewing possibility. The case report is interesting but there are some areas that should be additionally described. Please explain more clearly why you focused on the Caffey disease when there are other more common medical statuses that can be considered as the comparators to trauma. Fever, hyperostosis, elevated enzymes are not unique for Caffey disease. Please unify data from the line 131 and 185. In 130 you wrote no flat bone involvement and in 183 any bone can be involved. Please read fragments in this area and delete repeated data. Please re-write conclusions, because this part should describe work data not the principles from the student book. To obtain a detailed history and perform a thorough examination should not be the first conclusion in this work.

Author Response

Reviewer 1

Thank you for reviewing possibility. The case report is interesting but there are some areas that should be additionally described.

Thank you for reviewing our manuscript titled, "Differential Diagnosis between Child Abuse and Infantile Cortical Hyperostosis: A Case Report and Literature Review" and for giving us the opportunity to resubmit the paper. During the revision, we have tried to address all the concerns raised by you and the reviewers.

Please explain more clearly why you focused on the Caffey disease when there are other more common medical statuses that can be considered as the comparators to trauma. Fever, hyperostosis, elevated enzymes are not unique for Caffey disease.

We appreciate your important comment.

Since fever, elevated enzyme levels, and hyperostosis are not the only findings of Caffey disease, we considered several possibilities first. However, the most decisive was the report from the radiologic department. Radiology findings revealed that the bone involvement was diffuse and did not appear to be trauma and the diagnosis was Caffey's disease. In retrospect, it is thought that it was a misdiagnosis by the radiology department and that there was confusion about the diagnosis.  That is the reason we are reporting this case.

Please unify data from the line 131 and 185. In 130 you wrote no flat bone involvement and in 183 any bone can be involved. Please read fragments in this area and delete repeated data.

The sentence on line 131 is a sentence that states that, unlike in Caffey disease, there was no invasion of flat bone in our case. If there is any misunderstanding, I will correct it as follows.

Line 129

First, unlike the bone lesions of Caffey disease that characteristically involve the mandible and skull, no flat bone involvement was evident in our case. In Caffey disease, the mandible is most frequently involved (in 70–90% of the cases) and is often used as a defining diagnostic criterion

In addition, as you mentioned, we will match the prevalence data and revise the sentence as follows:

Line 184

Mandibular involvement is seen in 95% of the cases and can be useful to differentiate Caffey disease from child abuse.

=>Mandibular involvement is seen in 70-90% of the cases and can be useful to differentiate Caffey disease from child abuse.

Please re-write conclusions, because this part should describe work data not the principles from the student book. To obtain a detailed history and perform a thorough examination should not be the first conclusion in this work.

We appreciate your important suggestions and comment. We agree with your opinion. As you mentioned, we revised the conclusion.

Line 196

We report this case because child abuse cases may be confused with Caffey disease. This case report can therefore help distinguish between Caffey disease and child abuse.

Reviewer 2 Report

The Authors did A Case Report and Literature Review on:”Differential Diagnosis between Child Abuse and Infantile Cortical Hyperostosis”.

Well written but I have to comment on few things:

  1. In line 67 authors wrote: “The patient gave written consent for publication of this case report” The boy is an 8 months (11 months) infant. There is no way he gave his consent, and it is a bit hard to belive that accused parents would do so. And if they have any power over the child if this was an child abuse and child social service took place and gave a child to a legal guardian (but I don’t know this nor how is this managed in Republic of Korea).
  2. Line 77. Please revise the sentence! Rubbing and scratching can also be traumatic and are in fact traumatic movements (depending of force and duration)
  3. Line 88. Have to ask, cannot but wonder, authors did skeletal survey and suspect on child abuse only based on eye finding (and the child had fever), without any other signs, skin lesions and bruises? A bit farfetched, if you ask me... Nevertheless, it seems the suspicion was confirmed but I think that most of surgeons wouldn’t suspect it based on just that. Or there were some other implications? Parent mood? discrepancies in stories of the parents, weird behavior. Other diagnostic?

Author Response

Reviewer 2

Thank you for reviewing our manuscript titled, "Differential Diagnosis between Child Abuse and Infantile Cortical Hyperostosis: A Case Report and Literature Review" and for giving us the opportunity to resubmit the paper. During the revision, we have tried to address all the concerns raised by you and the reviewers.

Well written but I have to comment on few things:

  1. In line 67 authors wrote: “The patient gave written consent for publication of this case report” The boy is an 8 months (11 months) infant. There is no way he gave his consent, and it is a bit hard to belive that accused parents would do so. And if they have any power over the child if this was an child abuse and child social service took place and gave a child to a legal guardian (but I don’t know this nor how is this managed in Republic of Korea).

Thanks you your important question.

In this case, the medical staff considered child abuse and reported it to the police, but the evidence was insufficient. However, the patient's grandfather reported the parents as child abusers. The parents were isolated from the child and the patient was taken care of by the grandfather. The consent was obtained from the grandfather, the legal guardian of the patient.

  1. Line 77. Please revise the sentence! Rubbing and scratching can also be traumatic and are in fact traumatic movements (depending of force and duration)

We appreciate your comment.

We revised the sentence accordingly:

Line 77

“There was eschar on both canthal areas, which, according to his mother, was produced by rubbing but appeared to be more likely due to severe trauma.”

  1. Line 88. Have to ask, cannot but wonder, authors did skeletal survey and suspect on child abuse only based on eye finding (and the child had fever), without any other signs, skin lesions and bruises? A bit farfetched, if you ask me... Nevertheless, it seems the suspicion was confirmed but I think that most of surgeons wouldn’t suspect it based on just that. Or there were some other implications? Parent mood? discrepancies in stories of the parents, weird behavior. Other diagnostic?

We appreciate your important question.

In Korea, child abuse has emerged as an important social issue in recent years.

Therefore, it is a routine to conduct a skeletal survey in the case of a suspected patient in the emergency room. In addition, medical staff always keep in mind the possibility of child abuse in the case of traumatized children.

In fact, this patient started the examination with suspicion of her mother's remarks, However, when we saw the father taking care of the patient with sincerity in the inpatient ward, the suspicion of child abuse decreased. It seems that the diagnosis was more confused. In the end, it was concluded that it was parental child abuse; therefore, we learned that child abuse should not be excluded from the act of caring for a child.

Reviewer 3 Report

I was pleased  to read Dr. Du  Yeon  Lee et all's un well presented case report

Author Response

Thank you for reviewing our manuscript titled, "Differential Diagnosis between Child Abuse and Infantile Cortical Hyperostosis: A Case Report and Literature Review" and for giving us the opportunity to resubmit the paper. We appriciate your comment.

Reviewer 4 Report

The manuscript is a case report study that described a case of an 8-month-old boy presented with extensive periosteal reaction was observed in both upper and lower long bones. There was no specific history of injury. Caffey disease was initially considered as the diagnosis because the patient displayed fever and hyperostosis of multiple bones with elevated erythrocyte sedimentation rates, C-reactive protein, and alkaline phosphatase. However, we suspected child abuse based on the following clinical and radiologic features.

I read the article with interest, the title is well thought out and faithfully reflects the content of the study

The abstract is adequately developed, and it is useful to frame the purpose and the characteristics of the study.

In the introduction, the characteristics of child abuse have been described.  

The case description has been adequately developed.

 The discussion is sufficiently described.

Nevertheless, some minor changes are needed to be considered suitable for publication.

Comment 1: In the background: it would be appropriate to refer to the characteristics of the study.

Comment 2: In the background: Some information about, diagnosis, and treatment of child abuse should be deepened please add appropriate bibliographical references. (Pavone V et al, (2016) "Awareness and Recognition: The Importance of the Orthopaedist in Child Abuse").

Comment 3: In case description: Was the definitive diagnosis made by a pediatric orthopedist?

Comment 4: In case description: How did you manage multiple fractures?

Comment 5: In case description: Weren't abdominal and thoracic trauma also present?

Comment 6: In case description: From the images that appear to be present of the collarbone? have you excluded them?

Comment 7: In case description: How long did it take to re-evaluate the patient clinically and radiographically?

Comment 8: In case description: Did you hospitalize the patient to rule out other pathologies?

Comment 9: In the discussion: It would be appropriate to refer to previous studies carried out on the same topic, for example: (Kairys S. et al, (2020) "Child Abuse and Neglect")

Comment 10: Finally, additional English editing is needed. The Non-Native Speakers of English Editing Certificate was not signed.

Author Response

Reviewer 4

The manuscript is a case report study that described a case of an 8-month-old boy presented with extensive periosteal reaction was observed in both upper and lower long bones. There was no specific history of injury. Caffey disease was initially considered as the diagnosis because the patient displayed fever and hyperostosis of multiple bones with elevated erythrocyte sedimentation rates, C-reactive protein, and alkaline phosphatase. However, we suspected child abuse based on the following clinical and radiologic features.

I read the article with interest, the title is well thought out and faithfully reflects the content of the study

The abstract is adequately developed, and it is useful to frame the purpose and the characteristics of the study.

In the introduction, the characteristics of child abuse have been described.

The case description has been adequately developed.

 The discussion is sufficiently described.

Nevertheless, some minor changes are needed to be considered suitable for publication

Thank you for reviewing our manuscript titled, "Differential Diagnosis between Child Abuse and Infantile Cortical Hyperostosis: A Case Report and Literature Review" and for giving us the opportunity to resubmit the paper. During the revision, we have tried to address all the concerns raised by you and the reviewers.

Comment 1: In the background: it would be appropriate to refer to the characteristics of the study.

We appreciated your comment. I have added the sentence in the background that “This study was published to highlight the importance of the differential diagnosis between Caffey disease and child abuse” (Line 44)

Comment 2: In the background: Some information about, diagnosis, and treatment of child abuse should be deepened please add appropriate bibliographical references. (Pavone V et al, (2016) "Awareness and Recognition: The Importance of the Orthopaedist in Child Abuse").

Thank you for the suggestion. I will refer to the above paper to supplement the content of child abuse in the background.

Line 42

The important role of the orthopedic specialist is also emphasized in the mini review by Pavone et al.

Comment 3: In case description: Was the definitive diagnosis made by a pediatric orthopedist?

Yes, we were suspicious at first because of the diagnosis of Caffey disease in the radiology findings. However, the corresponding author, myself and colleagues diagnosed child abuse based on the pattern of the periosteal reaction and fracture pattern. However, legally, the grandfather of the patient reported the parents as child abusers. As the parents admitted to the abuse, the legal process proceeded.

Comment 4: In case description: How did you manage multiple fractures?

There were multiple fractures, but all were possible to manage with conservative treatment; therefore, conservative treatment through a simple splint was performed, and complete union and remodeling findings were observed after 3 months.

Comment 5: In case description: Weren't abdominal and thoracic trauma also present?

Rib fractures and abdominal injuries were also suspected; however, an expert in each area inspected the injuries but could not find any abnormalities.

Comment 6: In case description: From the images that appear to be present of the collarbone? have you excluded them?

Clavicle fractures were identified, but we thought that surgical treatment would not be necessary, and conservative treatment was performed. Union was confirmed 3 months later.

And three months later, the union was found.

Comment 7: In case description: How long did it take to re-evaluate the patient clinically and radiographically?

The patient is now isolated from his parents and is being treated in a shelter. The three-month follow-up was the last at our hospital. Currently, the patient is receiving regular ophthalmology and orthopedic treatment at other hospitals.

Comment 8: In case description: Did you hospitalize the patient to rule out other pathologies?

The patient first complained of discomfort in the eyes and reported the child abuse discovered during the evaluation. Surgery was performed on the eye and conservative treatment was performed for multiple fractures. For other abdominal and chest injuries, the experts of each area performed an examination but no specific abnormalities were found.

Comment 4-8

Based on your questions and advice, We add the following sentence to the discussion.

(Line193)

Although rib fractures and abdominal injuries were also suspected after expert inspections, no abnormalities were found. There were however multiple fractures, including clavicle fractures, that were identified, but all were possible to manage with conservative treatment. Conservative treatment using a simple splint was thus performed, and complete union and remodeling findings were observed after 3 months. The patient is now isolated from his parents and is being treated in a shelter. The three-month follow-up was the last at our hospital.

Comment 9: In the discussion: It would be appropriate to refer to previous studies carried out on the same topic, for example: (Kairys S. et al, (2020) "Child Abuse and Neglect")

We appreciate your comment and advice. Based on the study you recommended. the following sentence has been added with reference to the discussion. “The clinician’s role is not only in prevention and early detection but also in continuing to care for and manage the child and the family.”(Line 203)

Comment 10: Finally, additional English editing is needed. The Non-Native Speakers of English Editing Certificate was not signed

We appreciate your advice, Yes, I will submit the English proofreading certificate and ensure the manuscript undergoes a round of English proofreading again.

Round 2

Reviewer 1 Report

Manuscript can be accepted in present form but scientific soundness is low.

Reviewer 2 Report

The authors did and explained what was asked of them...